# Tissue-specific consequences of tag fusions on protein expression in transgenic mice

Gillian C. A. Taylor[1,2], Lewis Macdonald[1], Natalia A. Szulc[3], Evelina Gudauskaite[1], Brianda Hernandez Moran[1], Jennifer M. Brisbane[1], Molly Donald[1], Ella Taylor[4], Dejin Zheng[5,6], Bin Gu[5,6], Pleasantine Mill[1,7], Patricia L. Yeyati[1], Wojciech Pokrzywa[3], Claudia Ribeiro de Almeida[4], Andrew J. Wood[1,2]*

1 MRC Human Genetics Unit, Institute of Genetics and Cancer, University of Edinburgh, Edinburgh, United Kingdom, 2 MRC National Mouse Genetics Network, Degron Tagging Cluster, Mary Lyon Centre, Harwell, United Kingdom, 3 Laboratory of Protein Metabolism, International Institute of Molecular and Cell Biology in Warsaw, Warsaw, Poland, 4 Immunology Programme, The Babraham Institute, Babraham Research Campus, Cambridge, United Kingdom, 5 Department of Obstetrics, Gynecology and Reproductive Biology, Michigan State University, East Lansing, Michigan, United States of America, 6 Institute for Quantitative Health Science and Engineering, Michigan State University, East Lansing, Michigan, United States of America, 7 MRC National Mouse Genetics Network, Congenital Anomalies Cluster, Mary Lyon Centre, Harwell, United Kingdom

* Andrew.j.wood@ed.ac.uk

## Abstract

Genetic fusion of protein tags is widely used to study protein functions *in vivo*. It is well known that tag fusion can cause unwanted changes in protein stability, but whether this is an inherent property of the tagged protein, or can be influenced by the cell and tissue environment, is unclear. Using a series of genome edited mouse models, we show that tag-dependent changes in protein expression can vary across different primary cell and tissue contexts. In one case (*Ncaph2*), a C-terminal auxin-inducible degron fusion strongly increased protein stability in some tissues but decreased it in others. Destabilisation resulted from tissue-specific 'leakage' of the auxin-inducible degron, which depended on TIR1 expression, and occurred selectively in the small intestine where basal concentrations of auxin/ indole-3-acetic acid can reach levels that are sufficient to trigger protein degradation in cultured cells. Stabilisation occurred in post-mitotic cells via an endogenous degradation signal situated at the NCAPH2 C-terminus, which normally undergoes activation upon cell cycle exit, but is inactivated by C-terminal tag fusion. Our results highlight the underappreciated importance of cell and tissue environment in determining the consequences of tag fusions on protein expression, which may be particularly important in animal models that contain diverse cell types.

## Author summary

To better understand the function of specific proteins, researchers need ways to quantify their abundance, to visualise their movements, to understand what other

**Data availability statement:** All relevant data are within the manuscript and its Supporting Information file. Code used to generate PSI predictions is available at https://github.com/filipsPL/degronopedia-ml-psi.

**Funding:** AJW received funding from the Medical Research Council UK, grant number MC_PC_21040, and from an MRC Unit award. WP received funding from the National Science Centre, Poland, SONATA BIS grant number 2021/42/E/NZ1/00190. CRA received funding from the Wellcome Trust, fellowship number 220196/Z/20/Z, the BBSRC Institute Strategic Immunology Programme, grant number BB/Y006917/1, and the BBSRC Institute Development Grant BB/IDG2310/1. PM, PLY received funding from the Medical Research Council UK, grant number MR_Y015002_1. PM received funding from the European Research Council (ERC) under the European Union's Horizon 2020 research and innovation pro-gramme, grant agreement n°866355. PM, PLY received funding from the Medical Research Council UK, grant number MR_Y015002_1 and grant number MC_PC_21047. DZ and BG received funding from the National Institutes of Health/National Cancer Institute, USA. Grant number R37CA269076 and R03CA286646. The funders played no role in the study design, data collection, analysis, decision to publish or preparation of the manuscript.

**Competing interests:** The authors have declared that no competing interests exist.

molecules they interact with, and to ask what happens when they are removed. This is commonly achieved via 'protein tagging', which uses genetic engineering to fuse extra amino acids onto the natural protein sequence. These extra amino acids, or 'tags', confer new functionality, but can also disrupt normal protein expression and function which complicates interpretation of experimental data. This has been described previously, but the underlying mechanisms are often not understood. Moreover, tagging is increasingly used in genetically engineered animal models where tagged proteins are expressed in different cell types and tissue environments. It was previously unknown whether the consequences of tag fusion are universal, or dependent on the specific environment in which the protein is expressed. In this manuscript, we show that tag fusions often change protein expression to a degree that is highly dependent on cell and tissue context. We characterise two underlying mechanisms, providing information that should help others to design more effective transgenic strategies that better align with the '3R' principles for humane animal research.

## Introduction

Protein tags are polypeptides that can be genetically fused to proteins in order to confer new functionality [1]. Originally used to enhance protein purification and detection [2,3], tags now enable an increasingly wide range of applications in molecular and cell biology, including accurate quantification, super-resolution and live cell imaging, proximity labelling, and targeted protein degradation [4–10]. The application of protein tags has expanded beyond cellular models, with advances in genome editing facilitating their use in more complex systems such as organoids and genetically engineered animals [11–16], providing insights into protein behaviours at the tissue and whole organism level.

The ability to engineer genetic fusions via targeted knock-in at the endogenous genomic locus has been particularly transformative [17–20]. Endogenously tagged proteins are transcribed under the control of their native regulatory elements, preserving the gene's natural intron/exon structure. Compared to transgenes expressed from extrachromosomal DNA, or random genomic integration, endogenous tagging greatly increases the likelihood that physiological patterns of spatiotemporal expression will be retained by the tagged construct.

However, even with endogenous expression from genome-edited loci, tag fusions carry a significant risk of altering expression and/or function at the protein level. Tags can disrupt protein folding, cellular localisation and inter-molecular interactions leading to changes in stability & activity. These issues are well documented in cultured, rapidly growing cells (Cho et al., 2022; Fort et al., 2017; Gameiro et al.; Huh et al., 2003; Skube et al., 2010; Weill et al., 2018). However, in animal models, fusion proteins are exposed to a wide variety of different cellular contexts, of which the impact is currently unclear.

Tags are typically fused to the N- or C-terminus of the protein-of-interest, even though terminal fusion is often not necessary for tag function. Protein termini are favoured because they are often unstructured [21], which can reduce the likelihood of tag fusion severely disrupting protein folding. The availability of ORFeome collections and high throughput techniques for cloning and tagging [22–24] also facilitate rapid construction and testing of terminally tagged fusions.

In recent decades it has become clear that endogenous protein termini are enriched for short linear motifs known as 'degrons', which determine protein half-lives by mediating interactions with trans-regulatory components of the ubiquitin proteasome system (UPS) [25–31]. High throughput screens have systematically mapped degrons in rapidly growing mammalian cell lines such as HEK293T [29,32–34]. However, recent data suggest that the regulation of protein stability may be particularly important for proteome regulation in non-dividing cells [35]. The complete repertoire of degrons that regulate protein levels under the diverse physiological conditions found in mammalian tissues remains to be defined.

A key property of degrons is that they can be transplanted onto new proteins by genetic fusion, where they confer new regulatory properties. This has been exploited by synthetic biologists seeking more effective ways to control protein dosage in experimental systems [36–38]. Chemical-inducible degron tags such as the auxin-inducible degron (AID) are increasingly popular for studying protein function, because they allow for precise temporal control of protein dosage using small molecules [36,39–42].

Here, we make use of genome edited mouse models expressing tagged proteins and develop a simple internally controlled system to compare the steady-state expression of wildtype versus tagged proteins, under endogenous transcriptional regulation, across tissues. In three out of five models tested, we find that the degree to which tags disrupt expression is significantly influenced by tissue context. We go on to characterise two distinct mechanisms by which terminal tag fusions can cause undesirable tissue-specific changes in protein expression and suggest strategies to minimise this problem at the allele design stage.

## Results

In prior work, we used CRISPR-Cas9 editing and homology-directed repair to generate mice with tags fused at the endogenous loci encoding NCAPH and NCAPH2, the kleisin subunits of the two mammalian condensin complexes (*Ncaph^AID:Clover^* and *Ncaph2^AID:Clover^*, Fig 1A, [43]). Tags consisted of AID and the fluorescent protein Clover, fused in tandem via a short flexible linker peptide to the kleisin C-terminus. We performed genetic crosses to combine each tagged allele with a transgene encoding the auxin receptor Tir1 expressed from the *Rosa26* locus (*Rosa26^Tir1^*) to generate double transgenics, which enable rapid kleisin degradation to be induced both *ex vivo* and *in vivo* by exposure to auxin (indole-3-acetic acid). As reported previously [43], both fusion proteins were functional in absence of auxin. Homozygous tagged pups were born at normal Mendelian ratios whereas homozygous null mutations in either gene cause embryonic lethality with complete penetrance [44]. However more detailed examination of animals homozygous for the *Ncaph2^AID:Clover^* allele revealed evidence of subtle perturbations including reduced body weight and litter sizes. Quantification of protein levels suggested that the AID:Clover tag did not substantially alter steady-state protein expression levels, because tagged proteins were expressed at similar levels to untagged, and the presence of the Tir1 transgene caused only a minimal reduction in tagged protein levels (< 10% reduction). However, our previous study focused primarily on haematopoietic cells and tissues [43].

### Diverse effects of tags on protein abundance across mouse tissues

To further investigate the effect of C-terminal tags on NCAPH2 expression in mice, we probed immunoblots from tissue lysates from *Ncaph2^AID:Cloverl/+^; Rosa26^Tir1/Tir1^* mice with an antibody raised against endogenous mouse NCAPH2 protein [45]. By using heterozygous tissue and comparing the intensity of bands corresponding to tagged versus wildtype NCAPH2, we were able to quantify the relative levels of tagged versus wildtype protein, expressed from the endogenous chromosomal locus, across tissues (Fig 1B and 1C). Of note, the lowest molecular weight band on the western blot does not

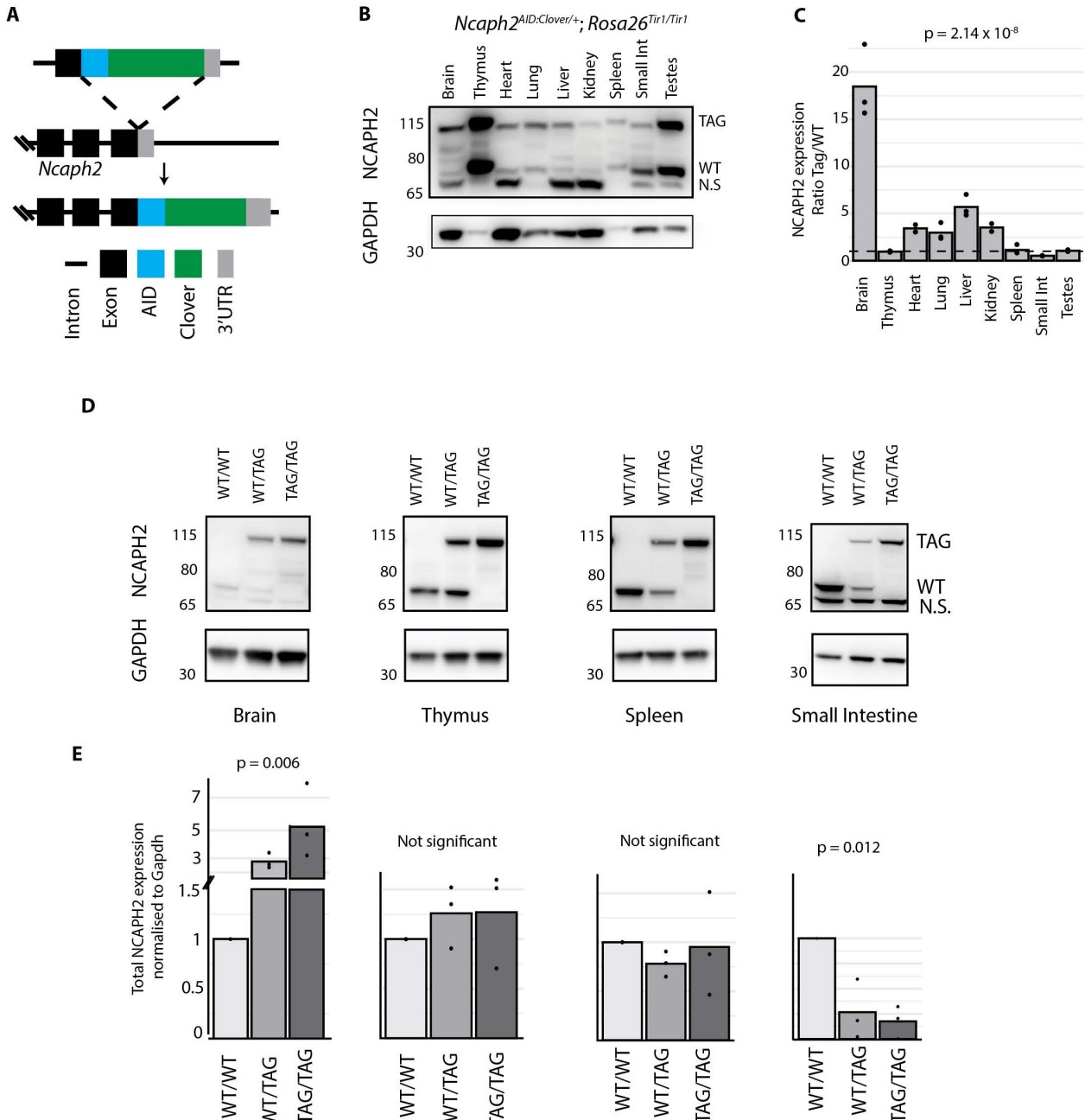

**Fig 1. C-terminal tagging of NCAPH2 can either increase or decrease protein expression.** A. Schematic illustration of the AID:Clover tag fusion at the endogenous *Ncaph2* C-terminus. B. Western blot of whole tissue lysates from *Ncaph2*^AID:Clover/+ *Rosa26*^Tir1/Tir1 mice, probed with an antibody raised against endogenous mouse NCAPH2. TAG and WT indicate allelic products of the *Ncaph2* gene, N.S. indicates a non-specific band (S1 Fig). GAPDH loading control is shown for reference but was not used for quantification purposes. C. Relative quantification of tagged versus wildtype protein based on band densitometry from western blots (n = 3 animals per tissue) D. Western blots show relative protein levels of NCAPH2 protein in tissues from *Rosa26*^Tir1/Tir1 animals with zero (WT/WT), one (WT/TAG) or two (TAG/TAG) AID:Clover alleles. E. Densitometric quantification of total NCAPH2 protein levels from panel D. The sum of WT and/or TAG bands was normalised to GAPDH in each case. p-values in panels C & E represent one-way ANOVA tests from n = 3 animals.

represent any known product of the *Ncaph2* gene [46] because it is unaffected by the presence of the tag (S1A Fig), by auxin-inducible degradation (S1B Fig) or knockdown using pooled siRNAs (S1C Fig).

Remarkably, the intensity ratio of tagged to wildtype bands varied greatly across tissues (Fig 1B and 1C). For example, the tagged protein was expressed at substantially higher levels than the untagged protein in the brain, at approximately even levels in the thymus and spleen, and at substantially lower levels in the small intestine (Fig 1C). This tissue-specific pattern was not a peculiarity of the heterozygous state, because mice homozygous for tagged alleles expressed higher total levels of NCAPH2 in the brain relative to wildtype, lower levels in the small intestine, and comparable levels in the thymus and spleen (Fig 1D).

To determine whether tag fusions exert context-dependent effects on other proteins, we examined protein levels in tissue samples from four additional genome edited mouse lines. In each case, western blots generated from heterozygous tissue lysates were probed with antibodies that recognised both tagged and untagged protein isoforms, and densitometry was performed to determine their relative abundance. The first line had an APEX2 tag (28 kDa) fused at the C-terminus of *Ift81*, encoding an intraflagellar transport protein. Bands of the expected size were observed in highly ciliated tissues (testes, trachea), where tagged and untagged protein isoforms were detected at approximately equal levels (S2A Fig and S2B Fig).

The next transgenic line had an AID tag (7kDa) N-terminally fused to the X-linked *Ddx3x* gene, encoding an RNA helicase with a broad expression pattern. Across tissues, the tagged protein was expressed at levels that were qualitatively lower than wildtype, but the small bandshift precluded quantitation and multi-tissue comparison (S2C Fig and S2D Fig). A third line (*Sox2^Halo^*) showed limited expression in adult tissues but was readily detected in brain and lung (Fig 2A). Whereas tagged protein was slightly (1.6 - fold) more abundant in lung tissue, it was substantially more abundant (~6-fold) in the brain (Fig 2B and 2C). For the fourth line (*Ncaph^AID:Clover^; Rosa26^Tir1^*) we used neonatal tissues due to the restricted expression profile in adults. The tagged protein was less abundant than the untagged across all tissues, but the expression ratio varied from almost equal (90% of untagged in the liver) to substantially lower (~30% of untagged in the small intestine (Fig 2D–2F)).

In summary, our data show that the impact of terminal tag fusions on protein expression can be highly dependent on tissue-context. For three out of five alleles tested (*Ncaph^AID:Clover^, Ncaph2^AID:Clover^, Sox2^Halo^*) tagged proteins were expressed at near-wildtype levels in some tissues, but significantly higher or lower levels in others.

### Auxin-inducible degron leakage is tissue-specific

For NCAPH2, the C-terminal AID:Clover tag fusion can influence expression via at least two distinct mechanisms: one that increases protein levels in tissues such as the brain, heart and liver, and another that reduces expression in the small intestine. Both mechanisms are post-transcriptional because tissue-specific differences in the ratio of tagged to wildtype gene product were not observed at the RNA level (S3A Fig). Similarly, the expression ratio of tagged to untagged mRNAs was not significantly different across tissues for the *Ncaph^AID:Clover^* or *Sox2^Halo^* lines (S3A Fig and S3B Fig).

To investigate the mechanisms underlying reduced expression of NCAPH2^AID:Clover^ protein in the small intestine, we considered the possibility of tissue-specific degradation leakage. This results in the downregulation of AID-tagged proteins in a TIR1-dependent manner in the absence of exogenous auxin. Degradation leakage of AID-tagged proteins has been widely reported in yeast, C. elegans and mammalian cells [20,47–50], but has not yet been observed in mouse models [43,51,52]. Although the underlying mechanisms are incompletely understood, we reasoned that leakage could occur when endogenous levels of the inducer molecule indole-3-acetic acid (IAA) exceed thresholds required for TIR1-dependent ubiquitination of AID-tagged proteins. Endogenous IAA concentrations vary between individuals due to factors such as diet, stress, and various disease processes. However they also vary between tissues, with particularly high levels in the gastrointestinal tract due to the conversion of tryptophan to IAA by commensal bacteria [53–55].

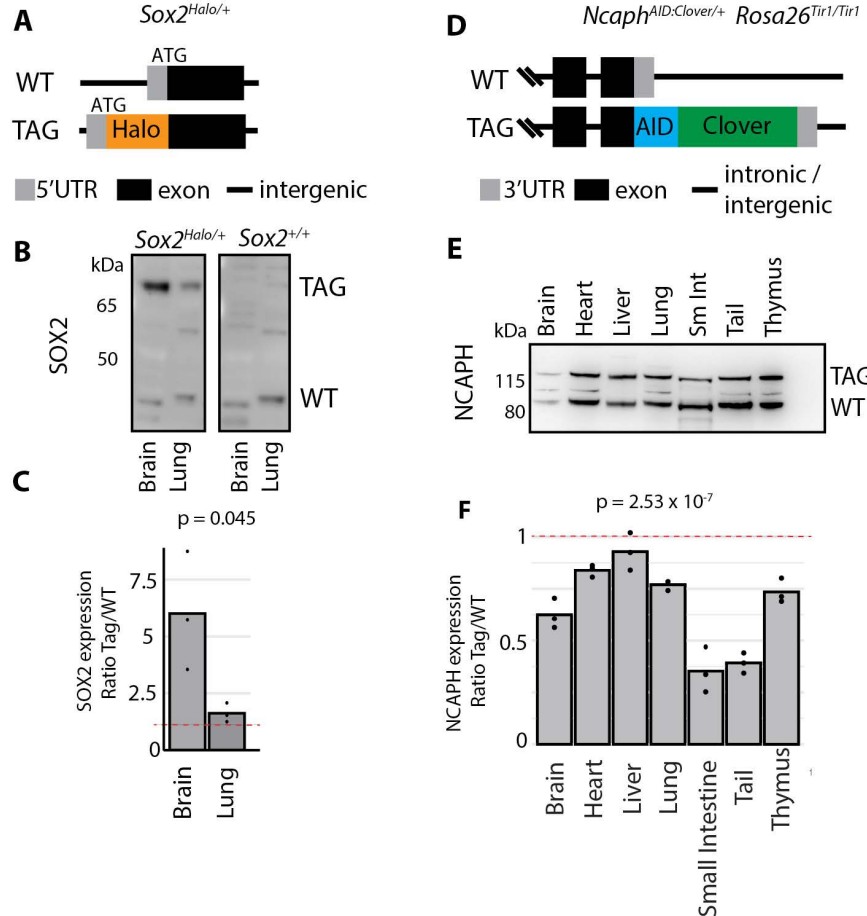

**Fig 2. Terminal tag fusions at SOX2 and NCAPH cause tissue-specific effects on protein expression.** A. Schematic illustration of the Halo-tag fusion at the endogenous SOX2 N-terminus. B. Western blot prepared from whole tissue lysates from *Sox2^Halo/+* and *Sox2^+/+* mice, probed with an antibody against endogenous SOX2. TAG and WT indicate allelic products of the *Sox2* gene. kDa indicates molecular weight markers. C. Relative quantification of tagged versus wildtype SOX2 protein based on band densitometry from western blots from heterozygous tissue. p-values represent unpaired t-test of difference between ratios, n = 3 animals. D. Schematic illustration of the AID:Clover fusion at the endogenous *Ncaph* C-terminus. E. Western blot from whole tissue lysates of *Ncaph^AID:Clover/+; Rosa26^Tir1/WT* mice, probed with an antibody raised against endogenous NCAPH. TAG and WT indicate allelic products of the *Ncaph* gene. F. Relative quantification of tagged versus wildtype protein based on band densitometry from western blots as shown in panel E. p-value represents one-way ANOVA test from n = 3 animals.

To determine whether basal IAA levels in the mouse small intestine could reach concentrations sufficient to induce degradation of AID-tagged proteins, we first mined published data to approximate molar concentrations in tissue homogenate from the mouse ileum [55]. This analysis revealed IAA concentrations in the range of 15–20µM in tissue from control animals that were not subject to experimental intervention (Fig 3A). In a proxy cell culture system, IAA concentrations within this range were sufficient to induce robust degradation of an AID-tagged GFP reporter following 24 hours of exposure (Fig 3B). *In vivo*, loss of TIR1 increased the expression of AID-tagged relative to untagged gene product in the small intestine (Fig 3C) but had minimal effect in tissues such as the thymus, where the ratio remained approximately 1:1 regardless of *Tir1* genotype (Fig 3D). The same tissue-specific effect of TIR1 loss was observed in a second AID-tagged line (*Ncaph^AID:Clover*); removal of TIR1 increased the ratio of tagged to untagged NCAPH protein in the small intestine (Fig 3E) but had minimal effect in the liver (Fig 3F). These data confirm that TIR1-dependent AID leakage is tissue-specific in mice, and disproportionately affects a tissue with high endogenous IAA production.

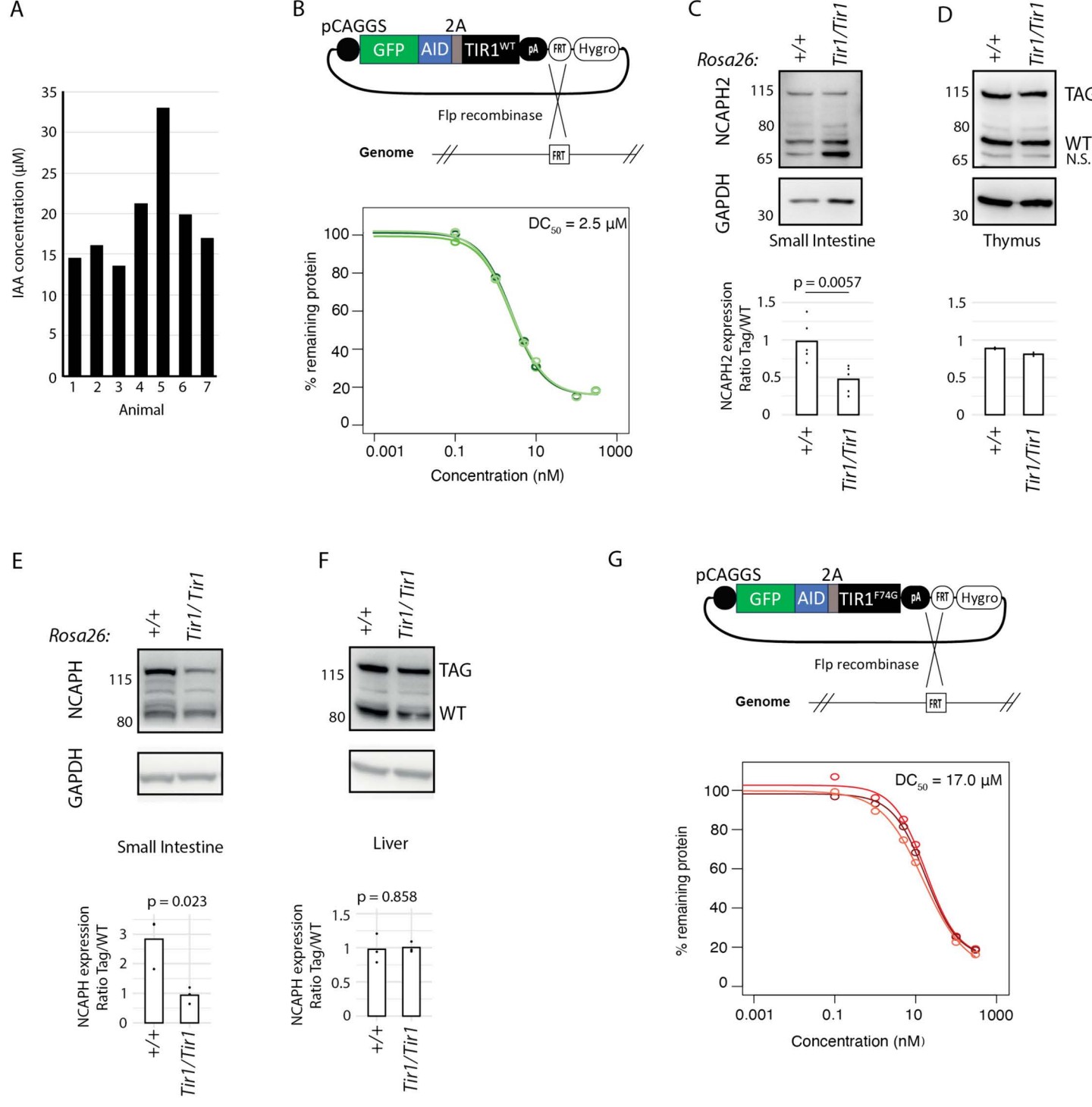

**Fig 3. Degradation leakage of the auxin-inducible degron is tissue-specific.** A. Endogenous indole-3-acetate (IAA) concentrations measured by targeted metabolomics (LC-MS) analysis of ileum tissue homogenate from untreated mice [55]. B. A GFP-AID reporter was integrated into a safe harbour locus in the genome of T-Rex HeLa cells using Flp/FRT recombination. The sensitivity of this reporter to degradation following 24 hour exposure to IAA was then tested in a dose response experiment, with GFP fluorescence measured by flow cytometry, background fluorescence subtracted and normalised to vehicle-only treated cells. Response curves from three biological replicate experiments are shown, and the concentration required for 50% of the maximum degradation response (DC$_{50}$) C & D. Western blots performed on tissue lysates from animals heterozygous for *Ncaph2*$^{AID:Clover}$ show

that the presence of *Tir1^{WT}* significantly reduces the expression of tagged (TAG) relative to untagged (WT) protein in the small intestine (panel C, n = 5 biological replicates) but has minimal effect in the thymus (panel D, n = 2). p-value represents two-tailed unpaired t-test. N.S. = non-specific band. E & F Western blots performed on tissue lysates from animals heterozygous for *Ncaph^{AID:Clover}* in the presence versus absence of TIR1 (n = 3). Data are presented as described for panels C & D. G. Sensitivity of a GFP-AID reporter to IAA-mediated degradation in combination with *Tir1^{F74G}*. Data are presented as described for panel B.

The recently developed AID2 system incorporates a missense mutation (F74G in *Tir1* from *Oryza sativa*) at the IAA binding surface of the TIR1 protein and has been reported to reduce TIR1-dependent leakage [39,49]. However, to our knowledge the relative sensitivity of TIR1^{WT} and TIR1^{F74G} to IAA-mediated degradation has not been precisely quantified. Using an isogenic HeLa cell system, we calculated that TIR1^{F74G} requires approximately 7-fold higher concentrations of IAA to achieve equivalent levels of GFP reporter protein degradation as TIR1^{WT} (Fig 3B and 3G). This provides both an explanation for the reduced degradation leakage in the AID2 system, and a clear rationale for using AID2/ TIR1^{F74G} in future *in vivo* work [39,56–58].

### C-terminal NCAPH2 tags disrupt proteasome-dependent turnover upon cell cycle exit

Tissues in which the C-terminal NCAPH2 tag fusion led to increased protein abundance (brain, heart, liver), tended to have a higher proportion of post-mitotic cells compared to tissues where protein levels were unaffected or decreased (spleen, thymus, small intestine, Fig 1). To investigate this further, we derived primary neural stem cells (NSCs) from the embryonic brains of mice heterozygous for the *Ncaph2^{AID:Clover}* allele (*Ncaph2^{AID:Clover/+}*; *Rosa26^{Tir1/Tir1}*, Fig 4A). These NSCs were maintained in the presence of epidermal growth factor (EGF) and fibroblast growth factor (FGF) [59]. In contrast to adult brain tissue, these cycling neural cells had approximately equal expression of tagged versus untagged protein (Fig 4B). To determine whether tags affected protein levels in a manner that depended on cell cycle exit, growth factors were withdrawn from the cell culture medium to induce quiescence (Fig 4C). In line with our hypothesis, the ratio of tagged to untagged protein increased as cells exited the cell cycle (Fig 4B). These data demonstrate that changes in cell state, such as those associated with differentiation or cell cycle status, can modify the degree to which protein tags impact steady-state protein expression.

Cell cycle exit correlated with an overall decrease in the level of both tagged and untagged NCAPH2 protein, as expected given that NCAPH2 functions primarily during mitosis [60,61], but the extent of downregulation was attenuated by the tag (Fig 4B and 4C). This suggested that the tag disrupts the mechanisms through which NCAPH2 is normally downregulated as cells stop dividing. Specifically, we found that the C-terminal tag interfered with proteasome-mediated turnover of NCAPH2 on cell cycle exit, because growth factor withdrawal in the presence of the proteasome inhibitor MG132 tended to reduce the preferential loss of wildtype protein over the tagged isoform (Fig 4D). However, this preferential loss of wildtype protein did not depend on the activity of Cullin Ring E3 ubiquitin ligases, as treatment with the neddylation inhibitor MLN4924 over the same time course failed to equalise expression of tagged and untagged protein isoforms (Fig 4E). Altogether, these data indicate that cell cycle exit in NSCs results in a proteasome-mediated downregulation of NCAPH2, and this process is disrupted by the presence of the C-terminal AID:Clover tag fusion.

### Terminal sequence duplication restores endogenous regulation of tagged proteins

It is well established that residues at the extreme N-termini of proteins often harbor natural degron motifs, which are regulated via the N-degron pathways [26,27]. More recently, it has become clear that protein C-termini are similarly enriched for degron motifs [29,32,33]. This led us to hypothesise that terminal tag fusion at NCAPH2 stabilised protein levels via a mechanism involving occlusion of a natural C-terminal degron. To test this, we modified the NCAPH2 C-terminus via CRISPR-Cas9 genome editing in wildtype E14 mouse embryonic stem cells (Fig 5A). We used two donor template molecules to generate distinct panels of genome edited clonal cell lines: one set heterozygous for a knock-in of eGFP

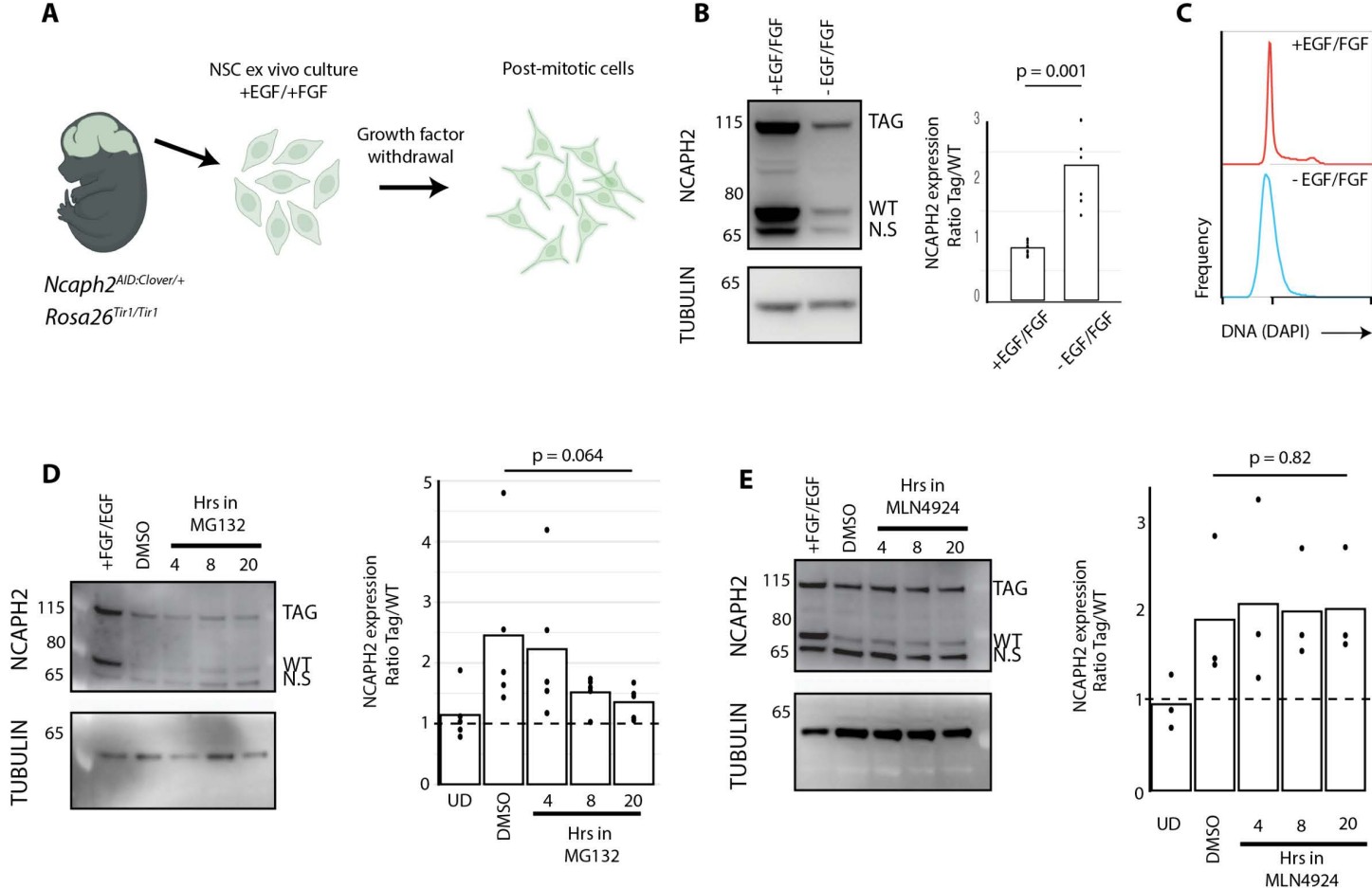

**Fig 4. C-terminal NCAPH2 fusions prevent proteasomal turnover following cell cycle exit.** A. Experimental schematic: Primary neural stem cells were isolated from embryonic brains (E12.5) and cultured in EGF and FGF. Growth factor withdrawal induces cell cycle exit. Created in BioRender. Wood, A. (2025) https://BioRender.com/17eaoyw. B. (left) western blots show changes in tagged and untagged NCAPH2 levels following 48 hours of growth factor withdrawal. (right) Western blot quantification. NCAPH2 band densities were expressed as a ratio of Tag/ WT. Tubulin loading control is shown for reference but was not used in the quantification. n = 9 replicates (undifferentiated) or 6 replicates (differentiated), each derived from the same original cell population, measured across at least two independent experiments. p value represents two-tailed unpaired-t-test. C. Flow cytometric quantification of DNA content confirms cell cycle exit following growth factor withdrawal. D. Changes in tagged and WT protein levels following growth factor withdrawal in the presence of the proteasome inhibitor MG132 (10 μM) for the specified time periods, or with vehicle for 20 hours. n = 5 replicates across two independent experiments. p value represents One way repeated measures ANOVA test. E. Changes in tagged and WT protein levels following growth factor withdrawal in the presence of the Cullin Ring E3 ligase inhibitor MLN4924 (5 μM), as described in panel D. p-value represents one-way repeated measures ANOVA test. Positive control experiments testing the activity of MG132 and MLN4924 were performed in parallel and are shown in S4 Fig.

(GFP only), and a second with heterozygous knock-in of eGFP plus a duplication of the thirty C-terminal amino acids from NCAPH2, to restore the endogenous C-terminus (GFP+30, Fig 5B and 5C). We then used flow cytometry to quantify fusion protein abundance either under rapid growth conditions (Serum+LIF - Fig 5D), or following 8 days of exposure to retinoic acid, which drives differentiation and proliferative arrest (Fig 5E).

During rapid growth, the level of eGFP expression was not significantly different between GFP only and GFP+30 cell lines (Fig 5D), consistent with C-terminal NCAPH2 residues not affecting protein levels in proliferating cells. In contrast, significant differences were observed following retinoic acid exposure: GFP+30 cells expressed the fusion protein at levels that were significantly lower, on average compared to GFPonly cells (Fig 5E). Altogether, these experiments support

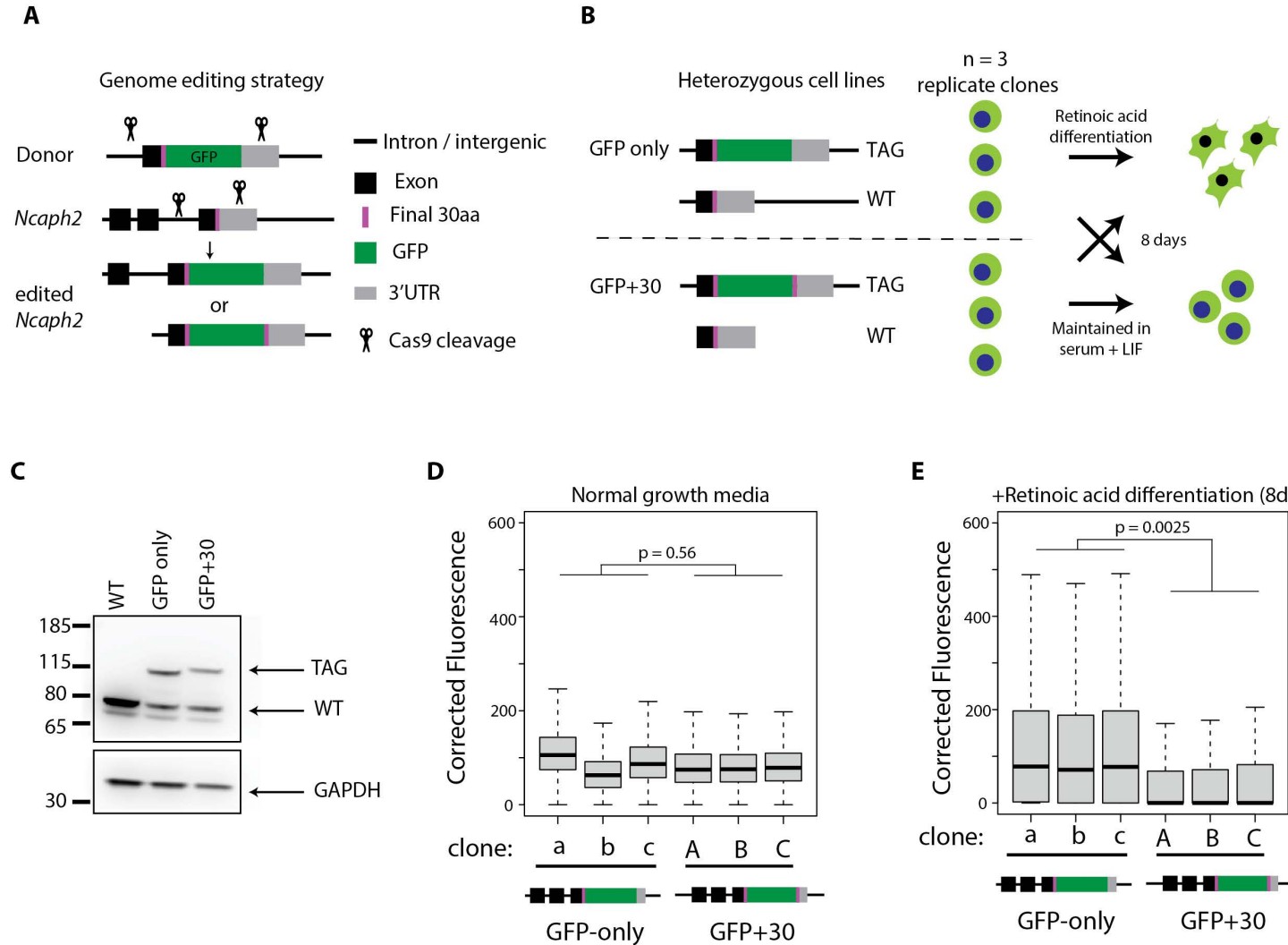

**Fig 5. Occlusion of a C-terminal NCAPH2 degron explains context-dependent effects of tag fusion.** A. Genome editing strategy to integrate a C-terminal GFP fusion at the endogenous *Ncaph2* locus in mouse embryonic stem cells. sgRNAs cut the genome in the terminal intron and 3'UTR of the endogenous *Ncaph2* gene, and an exogenous double stranded donor template, resulting in terminal exon replacement. Two different donors were used to integrate GFP only, or GFP together with a duplication of 30 C-terminal amino acids from endogenous NCAPH2 added to the tag C-terminus. B. Two panels of heterozygous cell lines were clonally expanded, then either maintained under conditions of self-renewal (serum+LIF) or differentiated for eight days in retinoic acid. C. Representative western blots confirm heterozygous expression of tagged *Ncaph2* alleles. D & E. Boxplots show the distribution of GFP fluorescence among >5000 single cells from each of the conditions shown in panel B. p-values represent two-tailed t-tests using the mean fluorescence value of all cells in a given condition.

the conclusion that the endogenous C-terminus of NCAPH2 contains a degron which enables proteasome-dependent turnover upon cell cycle exit and/or differentiation. By disrupting the function of this sequence, C-terminal tag fusions lead to increased expression of fusion proteins specifically in contexts where the degron is normally active.

## Discussion

Genetic tag fusions have great potential to elucidate protein functions in mammalian tissues and primary cells. Nonetheless, avoiding tag-dependent experimental artifacts remains a critical challenge in both cellular and animal systems. This

is of particular importance in animal models due to the '3R' principles; specifically, reducing the occurrence of failed experiments and minimising pathological phenotypes in animals expressing tagged alleles.

The overarching finding from this study is that genetic tag fusions can exert effects on protein abundance that are highly context-dependent, differing across tissues and cellular states. Below, we describe two tissue-specific mechanisms through which tags can cause undesirable changes in the expression of fusion proteins that differ in magnitude depending on trans-regulators in the local environment. It is important to note that transgenic mouse lines where tags cause severe expression perturbations typically either cannot be derived or are not maintained for practical and ethical reasons. This 'survivorship bias' means that our study likely underestimates the potential severity of tag-induced artifacts. However we believe that the mechanisms identified here, even in lines with relatively mild phenotypes, are likely to be relevant to other proteins and other tagging strategies, potentially with more severe consequences. For this reason, we also detail strategies through which the impact can be minimised during experimental design.

The first mechanism, observed in two transgenic lines (NCAPH:AID:Clover and NCAPH2:AID:Clover), destabilises AID-tagged fusion proteins in a Tir1-dependent manner specifically in tissues such as the small intestine, where the chemical inducer IAA is known to be produced at high levels by the microbiome. Intuitively this mechanism could apply to any AID-tagged protein, although precise levels of IAA sensitivity are likely to differ between fusions. Emerging data show that IAA concentration in mouse tissues can be affected by various conditions, including diet, stress, microbiome composition and disease states [53–55,62]. The approximately seven-fold reduced sensitivity of TIR1(F74G) to IAA (Fig 3) provides a mechanistic explanation for the reduced leakage that has been reported for the AID2 system [39,49] and supports the idea that this TIR1 variant should be preferred for applications in rodent models [39,56–58], and in other biological systems where IAA and structurally related molecules may occur at high levels in the local environment.

The second mechanism identified in this study results from inactivation of endogenous degrons by terminal tag fusion. This leads to enhanced fusion protein stability specifically under conditions where the degron is normally active. Fusion of a HiBiT tag was recently used to test for the presence of a degron at the C-terminus of FBXL15 in HEK293T cells [63]. Although tag fusion can serve as a useful demonstration of terminal degron activity, our strategy to restore degron function by duplicating terminal amino acids (Fig 5) provides an approach through which normal mechanisms of protein turnover can be maintained through careful allele design. Alternatively, tags can be fused at internal flexible linker regions situated away from protein termini [64–66], although the possibility of internal degrons or other protein interaction motifs at these sites should also be considered.

Is it possible to identify terminal degrons in silico, to determine whether certain protein termini should be avoided during allele design? The rules governing stability of protein N-termini are relatively well understood in cultured mammalian cells. Initiator methionine residues typically undergo proteolytic cleavage [67,68], and the amino acid immediately downstream strongly influences stability. Amino acids with small side chains (C/V/G/P/T/A/S) at this position tend to confer stability, whereas all other amino acids tend to confer instability at this position [25,26,30,68]. For example, SOX2, which was stabilised by an N-terminal HaloTag fusion (Fig 2B and 2C), normally has an unstable amino acid tyrosine at position 2, characteristic of a primary type II degron, whereas HaloTag has a stabilising alanine. In contrast, DDX3X was destabilised by the addition of an AID tag, even in the absence of TIR1 expression (S2D Fig). DDX3X normally has a stable serine at position 2, whereas the AID tag has an unstable lysine that is a predicted target of UBR-family E3 ligases.

The sequence determinants of degron function at protein C-termini, and at internal positions, are more complex. Nonetheless, numerous specific degron motifs have been identified and incorporated into searchable online resources [63,69–71]. The development of high throughput peptidomic screening methodology, particularly by the Elledge laboratory, has been transformative in this regard, allowing the stability of 23-mer peptides spanning the human proteome to be measured in a HEK293T cell system and leading to the identification of numerous novel degron motifs [29,31–34,68].

As part of the DEGRONOPEDIA web resource project (https://degronopedia.com, [63]), we developed a machine learning model trained on data from these high throughput screens [32,68]. This model was used to predict a Protein Stability

Index (PSI) score for N- and C-terminal 23-mer peptides for 15,037 human/mouse orthologs (S1 Table). An important caveat is that PSI scores are derived from data generated in HEK293T cells and therefore lack tissue-specific predictive power. Nonetheless, for tagging experiments where normal levels of protein expression need to be maintained (e.g., inducible degron tags), this resource, together with the online databases and seminal publications detailed above, will help researchers to avoid fusing tags at sites that perturb degron function at protein termini.

## Materials and methods

### Ethics statement

Animal work involving *Ncaph^AID:Clover^*, *Ncaph2^AID:Clover^* and *Ift-81^APEX2^* mice was approved by a University of Edinburgh internal ethics committee and was carried out under UK Home Office License numers PP2868089 and P18921CDE, in accordance with institutional guidelines under license by the UK Home Office. *Ddx3x^AID^* mice husbandry and experimentation was approved by the Babraham Institute's Animal Welfare and Ethical Review Body and complied with UK Home Office legislation. *Sox2^Halo^* mice husbandry was carried out under PROTO202000143 and PROTO 202300127, approved by the Michigan State University (MSU) Campus Animal Resources (CAR) and Institutional Animal Care and Use Committee (IACUC) in AAALAC credited facilities.

### Animal care and husbandry

Mice were maintained in a Specific Pathogen Free (SPF) environment in IVC cages at ambient temperature (22 – 25 C) under regular 12 hour light/dark cycles with free access to water and food (RM3 Chow, Special Diets Services). Tissues were harvested from young adults (between 6 and 20 weeks) unless otherwise stated. All *Ddx3x^AID^* animals were females; males and females were used interchangeably for other lines.

### Transgenic modifications

All transgenic mice had tags fused in frame at the endogenous chromosomal loci using CRISPR-Cas9 genome editing [11,13]. *Ncaph-* and *Ncaph2^AID:Clover^*; *Rosa26^Tir1^* mice are on a mixed C57BL/6J x CD1 background and have been described previously [43]. *Sox2^Halo^* mice are on a CD-1 background and have been described previously [11]. *Ddx3x^AID^* and *Ift81^Apex2^* are on a C57BL/6J background and will be described in detail in future publications. The amino acid sequences of all fusion proteins used in this study are listed in S1 Table.

### Tissue lysis

Young adult animals were humanely culled by cervical dislocation at 6 – 20 weeks post-partum unless otherwise stated. Tissues were harvested immediately, flash frozen in liquid nitrogen then stored in a -80 °C freezer. Between 10–30 mg of frozen tissue was weighed and homogenised in 1 mL lysis buffer (150 mM NaCl, 50 mM Tris-HCl pH 7.4, 2 mM EDTA, 1% SDS, protease inhibitors - Calbiochem Set V) for 30 seconds on ice using a T25 Ultra Turrax homogenizer. Cellular debris was pelleted via centrifugation at maximum speed for 15 minutes at 4 °C, and lysate was sonicated on ice until no longer viscous. Lysate was centrifuged again at maximum speed for 15 minutes at 4 °C. Supernatants were transferred into fresh tubes, and protein concentration was quantified using a Pierce BCA Protein Assay Kit (Thermo, 23228) following manufacturer's instructions.

NuPage LDS sample buffer (4X, Thermo, NP0007) and sample reducing agent (10x, Thermo, NP0009) was added to each sample prior to denaturation via heating at 80 °C for 10 minutes. Samples were used immediately or stored at -20 °C.

For small intestine samples, tissue was homogenised directly in 1X NuPage LDS sample buffer with sample reducing agent, denatured as described, then sonicated until no longer viscous.

## Western blotting

Denatured protein lysates (10 µg/sample) were loaded on to NuPAGE 4–12% Bis–Tris 1.0 mm Mini Protein Gels (Invitrogen, NP0321) alongside PageRuler Protein Ladder (5 µl/lane; Thermo Scientific, 26616) and run in 1×MOPS Buffer (Thermo, NP0001). Samples were typically run at 100 V for 90 min.

Transfers were performed by wet transfer. PVDF membranes were pre-soaked in 100% methanol (Fisher, 10284580) and rinsed briefly in Transfer Buffer (25 mM Tris [AnalaR, 103156X], 200 mM glycine [Fisher, G-0800–60], 20% methanol, 0.02% SDS [IGMM Technical Services]). Genie Blotter transfer device (Idea Scientific) was assembled with the gel and PVDF membrane placed between two layers of cellulose filter paper (Whatman, 3030–917) inside the loading tray. Once the apparatus was prepared, Transfer Buffer was filled to the top of the Genie Blotter and transfer proceeded for 90 min at 12 V.

Samples were blocked with 2% BSA (DDX3X) or 5% milk powder (all other blots) in Tris Buffered Saline with 0.1% Tween-20 with constant agitation, either at room temperature for 1 hr, or at 4°C overnight. Primary antibodies (S2 Table) were added to the corresponding block solution at the dilution shown. Membranes were incubated in the antibody dilutions with constant agitation, either at room temperature for 1 hr, or at 4°C overnight. Membranes were washed in TBS-Tween-20 solutions (0.1% Tween-20; three washes×10 min). HRP-conjugated secondary antibodies were also diluted in the corresponding block solution (with 0.1% Tween-20), and membranes were incubated with secondary antibody dilutions under constant agitation at room temperature for 1 hr. Membranes were then washed in TBS-Tween-20 solutions (0.1% Tween-20, four washes×10 min). Membranes were visualised on an ImageQuant4000 (Cytiva). Details of all antibodies used in this study are provided in S2 Table.

## Band densitometry

Band density was calculated using the Gel Analysis function of FIJI [72]. A rectangular selection was positioned around the lanes, and the Gel Analysis feature was used to draw a profile plot of the relative density of the contents of the selection. The peaks were enclosed with a line at the base, and the area of the peak was recorded. For calculation of band density ratios from heterozygous samples (e.g., Fig 1C), area measurements from two peaks within the same rectangular selection, at the correct size for tagged and wildtype protein were analysed. To calculate relative protein levels between wildtype, heterozygous and homozygous samples (Fig 1D), the ratio of peak area in wildtype samples versus loading control was set at 1, then equivalent ratios in heterozygous (combining both wildtype and tagged peaks) or homozygous samples were expressed relative to the wildtype value.

## Allele specific transcript quantification

For RNA extraction, adult tissue (brain, thymus, spleen, small intestine) were removed, snap frozen in liquid nitrogen, and stored at -80 °C until use. Between 10–30 mg of frozen tissue was weighed and homogenised in 1 mL Trizol (Invitrogen), and RNA was extracted following manufacturer instructions. Extracted RNA was DNase treated using DNAfree DNA removal kit (Invitrogen) prior to cDNA production.

2 mg total RNA was converted to cDNA using M-MLV Reverse Transcriptase, Random Primers, and RNasin (Promega) following manufacturer instructions.

Primer efficiency was tested for both primer pairs in RNA isolated from heterozygous spleen tissue across a series of 6 serial dilutions to ensure amplification of tagged and WT specific transcripts occurred at equal efficiency.

qRT-PCR was performed on a CFX96 Touch Real-Time PCR Detection System (BioRad) using SYBR Select Sybr Green Master Mix (Applied Biosystems). Primer Details are listed in S3 Table.

## Measurement of Indole-3-acetate in small intestine

Indole-3-acetate concentrations in the mouse ileum and colon were recently measured in an untargeted metabolomics study [55]. IAA concentrations in the ileum of control, untreated mice were extracted from S1 Table of that paper and

converted from ng/ml to µM. It is expected that IAA levels will vary based on numerous factors, including diet, microbiome composition, stress, and genetic background.

## Neural stem cell isolation and growth

Neural Stem Cells were derived from the telencephalon of individual E11.5-E13.5 embryos following a previously described protocol [59]. Once stably propagating, NSCs were cultured in T25 flasks, pre-coated in Laminin I (R&D Systems, 3446-005-01). When passaging, NSCs were washed with Wash Media (WM): DMEM/Ham's F-12 media with L-glutamine (Sigma-Aldrich, D8437-500), 300 mM D-(+)-glucose solution (Sigma-Aldrich, G8644), 1 × MEM Non-Essential Amino Acids Solution (Gibco, 11140050), 4.5 mM HEPES (Gibco, 15630056), 75 mg/ml BSA solution, 50 µM β-mercaptoethanol (Gibco, 31350–010), penicillin (70 mg/l, IGC Technical Services), and streptomycin (130 mg/l, IGC Technical Services), and propagated in Complete Media: WM supplemented with Epidermal Growth Factor (EGF) (Pepro-Tech, 315–09) and Recombinant Human FGF-basic (FGF) (PeproTech, 100-18B) each to final concentration of 10 ng/ml, 1 µg/ml Laminin (R&D Systems, 3446-005-01), 2.5 ml N-2 Supplement (100×) (Gibco, 17502048) and 5 ml B27 Supplement (50×) (Gibco, 17504044). Cells were cultured at 37°C, 5% $CO_2$ and passaged every 2–3 days.

To stimulate exit from cell cycle, cells were seeded in withdrawal media (WDM): complete media without addition of EGF and FGF, and cultured for 72 hours.

## Inhibition of the ubiquitin proteasome system

NSCs were seeded into 12 well plates at a density of 4x10^5 cells/well in WDM for 72 hours. Media was refreshed, and MG132 (10 µM) or MLN4924 (5 µM) or DMSO added for indicated time periods. Cells were lysed directly in lysis buffer (150 mM NaCl, 50mM Tris-HCl pH 7.4, 2 mM EDTA, 1% SDS, Protease inhibitors - Calbiochem Set V). Cellular debris was pelleted via centrifugation at maximum speed for 15 minutes at 4°C, and lysate was sonicated on ice until no longer viscous. Lysate was centrifuged again at maximum speed for 15 minutes at 4 °C and supernatants transferred into fresh tubes before SDS-Page analysis as described. For DNA content analysis, pellets were resuspended in Cytofix/Cytoperm solution (BD Bioscience, 554722) following the manufacturer's instructions and washed in Perm/Wash buffer (BD Bioscience 554723). Each sample was resuspended in 10 µg/mL DAPI solution in PBS for at least 60 min before data acquisition (BD LSRFortessa).

## siRNA

siGENOME mouse nCAPH2 siRNA SMARTpool (Dharmacon M-059491-01-0005) was transfected (75pmol/well in a 6 well plate) into 3T3 cells using lipofectamine 3000 (Invitrogen) according to manufacturer instructions. Cells were harvested and lysed in RIPA buffer (150mM NaCl, 1% NP-40, 0.5% Sodium Deoxycholate, 0.1% SDS, 50 mM Tris pH8.0) with protease inhibitors at 5x pelleted cell volume.

## ESC line generation

To generate *Ncaph2^GFPonly/+* and *Ncaph2^GFP+30/+* we used a non-homologous end-joining-based strategy with two gRNAs that cut either side of the final exon of the *Ncaph2* gene. gRNAs U1 and D3 targeting the upstream terminal intron and the downstream 3'UTR were cloned into px459v2 (addgene 62988) and sequence verified. Repair templates were manufactured by Twist Biosciences, and cloned into pCR-Blunt II-TOPO vector (ThermoFisher) using a Zero Blunt cloning kit (ThermoFisher) according to manufacturer instruction. Guides were transfected into E14 mESCs (see below) alongside repair templates in circular plasmid format using Lipofectamine 3000 (Invitrogen) according to manufacturer instructions. Colonies were selected with 1.3µg/mL puromycin for two days to select for positively transfected cells, then allowed to grow for approximately 1 week, before FACS sorting GFP positive cells (BD Aria or Cytoflex SRT. GFP positive cells were

seeded at low density, and individual colonies were picked into a 96 well plate, and screened by western blot for heterozygous colonies with tagged and WT bands at the correct size, indicating successful repair of one allele, and no deleterious mutations on the WT allele. A selection of positive colonies were then screened by sequencing. gDNA was produced using a DNeasy Blood and Tissue Kit (Qiagen) and the c-terminal region of Ncaph2 was amplified. The bands corresponding to WT or Tagged NCAPH2 were extracted with a Qiaex II Gel Extraction kit (Qiagen) and cloned into pGEM-T Easy before sequencing using SP6 and T7 primers. Nucleotide sequences used for gRNA cloning, donor template construction and genotyping are provided in S3 Table.

### ESC culture and differentiation

Embryonic Stem Cells (ESC) used are E14s, originally derived from 129/Ola. E14s were cultured (37 °C, 5% $CO_2$) in GMEM supplemented with 15% FCS, penicillin (70 mg/l, IGC Technical Services) and streptomycin (130 mg/l, IGC Technical Services), 2 mM L-glutamine (IGC Technical Services), 1 mM sodium pyruvate (Sigma-Aldrich, S8363), 50 µM β-mercaptoethanol (Gibco, 31350–010) and 1 × Non-Essential Amino Acids (Sigma-Aldrich, M7145), and 1,000 units/ml LIF (IGC Techical Services).

For retinoic acid differentiation, 0.5 x $10^6$ cells were seeded into a 6 well plate, into differentiation media (DM, as above, but without LIF). After 24 hours, DM was replaced with DM plus 5 µM Retinoic Acid (DM + RA) for day 1 of differentiation. The DM + RA media was refreshed daily for 8 days of differentiation, then cells were collected for FACS.

Cells were fixed in fix/perm (BD Bioscience) for 20 minutes on ice, then resuspended in PBS/DAPI (1 µM/mL), before data acquisition (BD LSR Fortessa).

### Generation and use of GFP:AID reporter HeLa cells

Cassettes encoding either GFP:mAID:2A:OsTir1[F74G], or GFP:mAID:2A:OsTir1[WT] were cloned into a pCAG expression vector containing a single FRT site. Full plasmid sequences are provided in S3 Table. These were transfected, together with a plasmid expressing Flp recombinase (pOG44) into TREx HeLa cells (Thermo Fisher) using Lipofectamine 3000 and the manufacturer's instructions, followed by hygromycin selection for approximately 2–3 weeks before FACS purification of GFP positive cells. Populations were >95% GFP+ at the time of seeding for experiments. For dose response experiments, cells were exposed to indole-3-acetic acid at the stated concentrations for 24 hours, then GFP levels were measured by flow cytometry (Fortessa, BD).

### Flow cytometry

E14 mouse embryonic stem cells or HeLa cells were gated according to forward and side scatter, then by DAPI height and area to exclude duplets. GFP signal scale values were exported and non-fluorescent background signal was subtracted from all values based on signal from parental cell lines lacking GFP transgenes, which were cultured, processed and analysed in parallel in the same way as their transgenic derivatives. Boxplots of corrected GFP values were produced in R using ggplot2. Dose response curves and DC50 values were produced in R using the package 'drc' [73].

### N-/C-terminal protein stability predictions

**Datasets.** The dataset of mouse protein-coding genes with one-to-one orthology to human genes was obtained from the Mouse Genome Informatics (MGI) database [74]. MGI identifiers (MGI IDs) and human gene nomenclature committee identifiers (HGNC IDs) were mapped to their corresponding UniProt accession numbers using UniProt's ID mapping service. Only reviewed entries were retained to ensure high-quality annotations. For mouse proteins, only those derived from *Mus musculus* were included, excluding entries from other mouse species. The corresponding protein sequences, gene names, and subcellular localization data were retrieved from UniProt. To ensure reliable PSI predictions, proteins

shorter than 24 amino acids were excluded from further analysis. In total, 15,421 mouse-human orthologous protein pairs met the inclusion criteria and were analyzed.

**Running predictions.** The Protein Stability Index (PSI) was predicted using a machine learning-based model (available at https://github.com/filipsPL/degronopedia-ml-psi) developed as part of the DEGRONOPEDIA web server [63]. This model was trained on experimentally determined N- and C-terminal stability data from Global Protein Stability (GPS) assays, which measured the stability of 23-residue terminal peptides across the human proteome [32,68]. Since initiator methionine removal occurs co-translationally in a manner dependent on the identity of the second amino acid and is linked to the Ac/N-degron pathway [75], PSI values were computed separately for methionine-retained and methionine-cleaved N-termini. The C-terminal PSI values were computed based on the last 23 amino acids of the protein sequence.

For each mouse-human orthologous protein pair, PSI values were recorded for both proteins to facilitate comparative assessment of terminal stability features. The final dataset, presented in S1 Table, includes original protein annotations, sequence lengths, and predicted PSI values. Full details of the machine learning approach, feature selection, and model training are described in [63].

## Supporting information

**S1 Fig. A ~65 KDa band detected by an anti-NCAPH2 antibody is not produced by the *Ncaph2* gene. A.** Western blot showing the bands detected by anti-NCAPH2 antibody from whole adult brain lysates of animals with 0, 1 or 2 alleles of the C-terminal AID:Clover tag fusion. N.S. indicates the position of non-specific band. **B.** Western blots on lysates of thymocytes from *Rosa26^Tir1/Tir1^* homozygotes either homozygous (Tg/Tg) or wildtype (+/+) for the *Ncaph2^AID:Clover^* allele cultured *ex vivo* for 3 hours in the presence or absence of 500 µM indole-3-acetic acid (IAA). **C.** Western blots of NIH-3T3 cell lysates, harvested 48 hours following transfection with an siRNA pool directed against mouse *Ncaph2*. Intensities of wildtype and non-specific bands were quantified relative to GAPDH over n = 2 independent transfections.
(PDF)

**S2 Fig. Multi-tissue expression of tagged DDX3X and IFT81 proteins. A.** Schematic showing the C-terminal APEX2 fusion at *Ift81*. **B.** Western blots on whole tissue lysates from adult animals heterozygous (left) or wildtype (right) for *Ift81^APEX2^*. The tag causes a band shift of 28 kDa. The protein is expressed at highest levels in motile ciliated cells which are abundant in the testes and trachea, where the ratio is approximately 1:1. At longer exposure times multiple bands of variable sizes were observed in wildtype and heterozygous samples from other tissues. Since there are no described iso-forms for IFT81, we assume these are non-specific bands. **C.** Schematic showing the N-terminal auxin-inducible degron (AID) fusion at *Ddx3x*. **D.** Western blots on tissue lysates from adult female mice heterozygous for *Ddx3x^AID^*. These mice did not co-express the TIR1 protein. Wildtype DDX3X is 73 kDa, and the AID tag causes a band shift of 7 kDa, which was insufficient to enable reliable individual quantification of tagged and wildtype bands. Western blot is representative of n = 3 biological replicates.
(PDF)

**S3 Fig. The ratio of tagged to untagged mRNA is not significantly different across tissues for *Ncaph2, Ncaph* and *Sox2*. A.** Schematic illustrating the design of allele-specific qRT-PCR assays to measure transcript ratios in tissues from *Ncaph2^AID:Clover/+^* and *Ncaph^AID:Clover^* (left) and *Sox2^Halo^* (right). **B.** Histogram showing ΔΔCt values, expressed as a ratio of tagged/ wildtype gene product from n = 3 biological replicate samples per tissue for *Ncaph2* and *Sox2*, n = 4 samples for *Ncaph*.
(PDF)

**S4 Fig. Control experiments show efficacy of MG132 and MLN4924 compounds.** Boxplots show degradation of NCAPH2^AID:Clover^ induced by 2 hour exposure of primary thymocytes to 100µM IAA. Degradation was inhibited by addition

of either 10μM MG132 (**A**) or 5μM MLN4924 (**B**), indicating successful inhibition of the proteasome, or Cullin RING Ubiquitin ligases, respectively.
(PDF)

**S1 Table. Mouse_Human_Ortholog_PSI. Predicted Protein Stability Index (PSI) values for N- and C-Termini of mouse-human orthologs.**
(XLSX)

**S2 Table. Antibodies used in this study.**
(XLSX)

**S3 Table. Nucleic acid sequences used in this study.**
(XLSX)

**S1 File. Uncropped western blots.** Image files for western blots used for protein quantification throughout the manuscript.
(PDF)

## Acknowledgments

We thank the Biological Research Facility at the Western General Hospital, University of Edinburgh, for animal husbandry, the University of Edinburgh Transgenic Service for generating *Ncaph*, *Ncaph2* and *Ift81* transgenic lines, and the IGC Flow Cytometry Facility. We are grateful to Dr Asif Nakhuda at the Babraham Gene Targeting facility for generating the *Ddx3x$^{AID}$* line, and to Dr Ben Davis for critical reading of the manuscript. This research was performed thanks to the IIMCB IN-MOL-CELL Infrastructure funded by the European Union – NextGenerationEU under National Recovery and Resilience Plan. IN-MOL-CELL Infrastructure was also funded by the European Union under Horizon Europe (Project 101059801 - RACE) and by RACE-PRIME project carried out within the IRAP programme of the Foundation for Polish Science co-financed by the European Union under the European Funds for Smart Economy 2021–2027 (FENG).

## Author contributions

**Conceptualization:** Gillian C. A. Taylor, Andrew J. Wood.

**Data curation:** Gillian C. A. Taylor, Andrew J. Wood.

**Formal analysis:** Gillian C. A. Taylor, Natalia A. Szulc, Evelina Gudauskaite, Andrew J. Wood.

**Funding acquisition:** Pleasantine Mill, Bin Gu, Wojciech Pokrzywa, Claudia Ribeiro de Almeida, Andrew J. Wood.

**Investigation:** Gillian C. A. Taylor, Lewis Macdonald, Evelina Gudauskaite, Brianda Hernandez Moran, Molly Donald, Ella Taylor.

**Methodology:** Gillian C. A. Taylor, Lewis Macdonald, Natalia A. Szulc, Wojciech Pokrzywa, Brianda Hernandez Moran.

**Project administration:** Andrew J. Wood.

**Resources:** Lewis Macdonald, Natalia A. Szulc, Jennifer m Brisbane, Ella Taylor, Dejin Zheng, Bin Gu, Pleasantine Mill, Patricia Yeyati, Wojciech Pokrzywa, Claudia Ribeiro de Almeida.

**Software:** Natalia A. Szulc, Wojciech Pokrzywa.

**Supervision:** Pleasantine Mill, Wojciech Pokrzywa, Claudia Ribeiro de Almeida, Andrew J. Wood.

**Visualization:** Andrew J. Wood.

**Writing – original draft:** Andrew J. Wood.

**Writing – review & editing:** Gillian C. A. Taylor, Natalia A. Szulc, Pleasantine Mill, Patricia Yeyati, Wojciech Pokrzywa, Claudia Ribeiro de Almeida, Andrew J. Wood.

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
