## [Decision Letter · Decision Letter 0]

13 Jun 2025

PGENETICS-D-25-00446

Tissue-specific consequences of tag fusions on protein expression in transgenic mice

PLOS Genetics

Dear Dr. Wood,

Thank you for submitting your manuscript to PLOS Genetics. After careful consideration, we feel that it has merit but does not fully meet PLOS Genetics's publication criteria as it currently stands. Therefore, we invite you to submit a revised version of the manuscript that addresses the points raised during the review process. In particular, addressing the questions raised by Reviewers 2 and 3 regarding mRNA isoforms and their expression levels will be important. 

Please submit your revised manuscript within 30 days Jul 13 2025 11:59PM. If you will need more time than this to complete your revisions, please reply to this message or contact the journal office at plosgenetics@plos.org. Please include the following items when submitting your revised manuscript:

We look forward to receiving your revised manuscript.

Kind regards,

Bruce A. Hamilton

Academic Editor

PLOS Genetics

Quanjiang Ji

Section Editor

PLOS Genetics

Aimée Dudley

Editor-in-Chief

PLOS Genetics

Anne Goriely

Editor-in-Chief

PLOS Genetics

**Journal Requirements:**

At this stage, the following Authors/Authors require contributions: Gillian CA Taylor, Lewis Macdonald, Natalia A Szulc, Evelina Gudauskaite, Brianda Hernandez Moran, Jennifer m Brisbane, Molly Donald, Ella Taylor, Dejin Zheng, Bin Gu, Pleasantine Mill, Patricia Yeyati, Wojciech Pokrzywa, Claudia Ribeiro de Almeida, and Andrew J. Wood. Please ensure that the full contributions of each author are acknowledged in the "Add/Edit/Remove Authors" section of our submission form.

The list of CRediT author contributions may be found here: https://journals.plos.org/plosgenetics/s/authorship#loc-author-contributions

https://journals.plos.org/plosgenetics/s/submission-guidelines#loc-parts-of-a-submission

5) We notice that your supplementary Figures are included in the manuscript file. Please remove them and upload them with the file type 'Supporting Information'. Please ensure that each Supporting Information file has a legend listed in the manuscript after the references list.

Potential Copyright Issues:

- Figure 4A. Please confirm whether you drew the images / clip-art within the figure panels by hand. If you did not draw the images, please provide (a) a link to the source of the images or icons and their license / terms of use; or (b) written permission from the copyright holder to publish the images or icons under our CC BY 4.0 license. Alternatively, you may replace the images with open source alternatives. See these open source resources you may use to replace images / clip-art:

7) Please amend your detailed Financial Disclosure statement. This is published with the article. It must therefore be completed in full sentences and contain the exact wording you wish to be published. Please ensure that the funders and grant numbers match between the Financial Disclosure field and the Funding Information tab in your submission form. Note that the funders must be provided in the same order in both places as well.

**Reviewers' comments:**

Reviewer's Responses to Questions

**Comments to the Authors:**

Reviewer #1: Review of "Tissue-specific consequences of tag fusions on protein expression in transgenic mice"

Manuscript Number: PGENETICS-D-25-00446

Overall Assessment: This manuscript by Taylor et al. investigates the often-underappreciated impact of protein tag fusions on protein expression levels in a tissue-specific manner within transgenic mice. The authors compellingly demonstrate that the consequences of tagging a protein can vary significantly across different tissues and cellular contexts, moving beyond observations typically made in cultured cell lines. They identify two distinct molecular mechanisms underlying these context-dependent effects: tissue-specific "leakage" of the auxin-inducible degron (AID) system due to endogenous auxin levels (particularly in the small intestine), and the disruption of an endogenous C-terminal degron leading to protein stabilization in post-mitotic cells (exemplified by Ncaph2). The study is well-designed, employing heterozygous animals for internal controls and providing robust biochemical evidence. The findings are significant for the broad community of researchers using tagged proteins in vivo, offering valuable insights for experimental design and data interpretation, and suggesting strategies to mitigate these issues.

Main Claims and Significance:

• Main Claims:

1. The effect of tag fusions on protein expression levels can be highly tissue-specific in transgenic mice.

2. C-terminal fusion of an AID:Clover tag to Ncaph2 can lead to increased protein stability in some tissues (e.g., brain) and decreased stability in others (e.g., small intestine).

3. Decreased expression of AID-tagged proteins in the small intestine is due to TIR1-dependent "leakage" of the degron system, likely triggered by basal levels of endogenous auxin/indole-3-acetic acid (IAA) in this tissue.

4. Increased stability of C-terminally tagged Ncaph2 in post-mitotic cells results from the inactivation of an endogenous degradation signal located at the Ncaph2 C-terminus, which is normally active upon cell cycle exit.

5. The authors propose strategies to mitigate these issues, such as using the AID2 system (with Tir1F74G) to reduce leakage and duplicating the endogenous protein terminus to preserve normal degradation signals.

• Significance: These claims are highly significant for the discipline. Genetic tagging is a cornerstone of modern molecular biology, and this work provides crucial caveats and practical guidance for its application in complex animal models. Understanding and mitigating tissue-specific artifacts is vital for the accurate interpretation of protein function studies in vivo and for upholding the 3Rs principles in animal research. The elucidation of specific mechanisms offers a rational basis for improved experimental design.

Novelty and Literature Context:

• Novelty: While it is known that tags can alter protein stability, this study's strength lies in its systematic demonstration and mechanistic investigation of tissue-specific variability in these effects within mouse models. The clear demonstration of endogenous IAA levels causing AID system leakage in a specific tissue (small intestine) in mice is a novel and important finding for users of this popular system. Similarly, the identification of a C-terminal degron in Ncaph2 whose function is masked by tagging and is active in post-mitotic cells, along with the proposed solution of terminal duplication, adds a new dimension to understanding tag interference.

• Literature Context: The authors have appropriately placed their claims within the context of previous literature. They acknowledge prior work on tag-induced protein alterations, the utility of endogenous tagging, the existence of N- and C-terminal degrons, and the development of inducible degron systems. The discussion fairly integrates their findings with existing knowledge, such as reports of AID leakage in other systems and the benefits of the AID2 system.

Data and Analyses:

The data and analyses presented generally provide strong support for the claims:

• Figures 1 and 2 clearly illustrate tissue-specific differences in tagged versus untagged protein ratios for Ncaph2, Sox2, and Ncaph using Western blotting in heterozygous animals.

• Figure 3 robustly supports the AID leakage mechanism in the small intestine by:

o Showing physiologically relevant IAA concentrations in the mouse ileum (from cited data).

o Demonstrating IAA-induced degradation of a reporter in cultured cells at similar concentrations.

o Showing Tir1-dependence of reduced tagged protein levels in the small intestine for both Ncaph2AID:Clover and NcaphAID:Clover.

o Quantifying the reduced IAA sensitivity of the Tir1F74G variant.

• Figure 4 supports the C-terminal degron hypothesis for Ncaph2 by showing:

o Increased tagged/wildtype protein ratio upon cell cycle exit in NSCs.

o Attenuation of wildtype Ncaph2 downregulation by proteasome inhibition (MG132).

• Figure 5 directly tests the C-terminal degron hypothesis by showing that duplicating the terminal 30 amino acids of Ncaph2 after a GFP tag restores its downregulation upon differentiation in mESCs.

• The allele-specific transcript quantification (Figure S3) effectively rules out transcriptional effects as the primary cause of the observed protein level differences.

• The use of heterozygous animals to provide internal controls for protein expression is a strong feature of the study design.

Suggestions for Improvement/Additional Work: The study is comprehensive and the conclusions are well-supported.

• While the authors discuss the potential for N-terminal degrons to be affected by tagging (Figure 2, SOX2 example, and discussion ), the mechanistic investigation focuses more heavily on the C-terminal Ncaph2 example and AID leakage. This is perfectly acceptable given the scope, but perhaps a brief mention in the discussion regarding the challenges of predicting or identifying internal degrons affected by tags (if tags were to be placed internally as a suggested solution ) could be considered.

Protocols and Data Availability:

• Protocols: The Materials and Methods section is exceptionally detailed and provides sufficient information for the experiments to be reproduced. This includes animal husbandry, generation of transgenic lines, tissue lysis, Western blotting, densitometry, cell culture (NSCs, ESCs, HeLa), molecular cloning, siRNA experiments, and bioinformatics for PSI predictions. The inclusion of specific reagents, concentrations, and equipment details is commendable. [e.g., 178, 181, 182, 184, 206, 212, 213, 217, 227, 228, 232, 238, 240, 252] The provision of Supplementary Tables for antibodies, nucleic acid sequences, and PSI predictions is excellent. Uncropped Western blots are also available.

• Data Availability: Original data appear to be within the manuscript and its Supporting Information files. Accession numbers for genes/proteins are implicitly handled via standard nomenclature and the PSI prediction pipeline uses UniProt. The software for PSI prediction is available on GitHub. Mouse lines are either previously described or noted for future detailed publication.

Adherence to Guidelines:

• The study appears to conform to relevant animal ethics guidelines, with approvals cited.

• The manuscript details should allow for robust assessment of the methodology.

Methodology Sufficiency for Reproduction: Yes, the methodological details provided are extensive and should allow other researchers to reproduce the experiments.

Software Availability: Yes, the machine learning model for PSI prediction is freely available on GitHub.

Organization and Clarity:

• The manuscript is very well organized, with a logical flow from introduction of the problem, presentation of results identifying tissue-specific effects, elucidation of underlying mechanisms, and finally, proposed solutions and broader implications.

• The writing is exceptionally clear, concise, and accessible even to those who may not be specialists in protein degradation or mouse genetics.

• Figures are generally clear and well-annotated, supporting the textual descriptions effectively.

Nomenclature and Abbreviations: Standard scientific nomenclature and abbreviations are used throughout the manuscript. Abbreviations are typically defined at their first use (e.g., AID, IAA, NSC).

Recommendation: This is a high-quality study presenting novel and significant findings that are of broad interest to the PLOS Genetics readership, particularly those working with transgenic models and protein tagging. The conclusions are strongly supported by rigorous experimental work. I recommend this manuscript for publication.

Reviewer #2: The manuscript by Taylor et al. presents a case study on the protein stability of tag-fused proteins in mice, with a particular focus on the condensin subunit NCAPH2. The authors previously reported a mouse line expressing NCAPH2-AID-Clover and found that the expression level of this protein was unaffected in hematopoietic cells (Macdonald et al., eLife, 2022). However, they observed that in some tissues, the expression levels were either higher or lower compared to the wild-type NCAPH2 protein (Fig. 1). Similarly, altered protein levels were detected when AID or HaloTag was fused to other proteins—such as DDX3, SOX2, and NCAPH—in a tissue-specific manner (Figs. 2 and S2). Based on these findings, the authors concluded that tag fusion can affect protein stability in a tissue-specific manner.

Regarding the underlying mechanism, the authors showed that NCAPH2-AID-Clover was degraded in the small intestine due to leaky degradation induced by endogenous indole-3-acetic acid (IAA) produced by gut microbes (Fig. 3). In contrast, in non-cycling neuronal cells in the brain, NCAPH2-AID-Clover levels were elevated, likely due to inefficient degradation of NCAPH2 in non-cycling cells (Fig. 4). To confirm whether the C-terminus of NCAPH2 contains a degron responsible for its reduced stability in non-cycling cells, the authors used a model ESC line expressing NCAPH2-GFP and found that the last 30 amino acids confer degron activity (Fig. 5).

Overall, this study was carefully conducted, and the presented data are convincing. Considering that tag-fusion strategies will be more frequently used in future studies with mice, this paper will be informative to anyone generating mice expressing a tag-fused protein.

Throughout the manuscript, the authors primarily focus on protein stability to explain the tissue-specific differences in expression. However, protein levels can also be influenced at the mRNA level, which the authors did not thoroughly investigate, except in the case of NCAPH2 (Fig. S3). Therefore, they should examine the mRNA levels of IFT81, DDX3, SOX2, and NCAPH from both the WT and tag-fused alleles (Fig. 2 and S2).

In Fig. 3C, the WB data were not ideal and difficult to interpret due to unequal protein loading. If the authors can show better blots, they should be presented.

Reviewer #3: The manuscript by Taylor et al. describes a detailed exploration of the tissue-specific effectiveness and artifactual background in transgenic tagging of proteins across in vivo tissues. Although the theoretical possibility of such artifacts are well-understood, the careful empirical description of the propensity of these issues with C- and N-terminal tagging as described here will provide a highly useful illustration to the scientific community. In general I find the experiments well-described and carefully considered to address the key point of the manuscript.

My only suggestion would be to incorporate a bit more analysis in Fig S1, as as the authors note the tissue-specific presence of that non-specific band does make the interpretation of Fig 1 somewhat difficult. I think Fig. S1 is reasonable, but particularly as deep tissue RNA-seq data is available (and that band seems non-tag dependent), it should be easy to check that there isn’t a unique short isoform of NCAPH2 that might also explain this band (i.e., is expressed in heart/liver/kidney but not thymus / lung / spleen)? Particularly as a non-expected alternative splicing or alternative polyA site isoform would show similar patterns in Fig S1 (would lose expression off the 3’ tag, wouldn’t be altered by tag presence or absence, and might show decreased efficiency of siRNA knockdown depending on siRNA location – particularly as the siRNA pool only shows ~30% knockdown efficiency anyway).

**Have all data underlying the figures and results presented in the manuscript been provided?**

Reviewer #1: Yes

Reviewer #2: Yes

Reviewer #3: Yes

PLOS authors have the option to publish the peer review history of their article (what does this mean? ). If published, this will include your full peer review and any attached files.

**Do you want your identity to be public for this peer review?** For information about this choice, including consent withdrawal, please see our Privacy Policy .

Reviewer #1: No

Reviewer #2: No

Reviewer #3: No

**Figure resubmission:**
---

## [Decision Letter · Decision Letter 1]

5 Aug 2025

Dear Dr Wood,

We are pleased to inform you that your manuscript entitled "Tissue-specific consequences of tag fusions on protein expression in transgenic mice" has been editorially accepted for publication in PLOS Genetics. Congratulations!

Yours sincerely,

Bruce A. Hamilton

Academic Editor

PLOS Genetics

Quanjiang Ji

Section Editor

PLOS Genetics

Aimée Dudley

Editor-in-Chief

PLOS Genetics

Anne Goriely

Editor-in-Chief

PLOS Genetics

Comments from the reviewers (if applicable):

Reviewer's Responses to Questions

**Comments to the Authors:**

Reviewer #2: The authors have appropriately revised the manuscript and addressed my previous concerns. I see no issues with its publication in PLOS Genetics.

Reviewer #3: The authors have addressed my concerns and I support publication

**Have all data underlying the figures and results presented in the manuscript been provided?**

Reviewer #2: Yes

Reviewer #3: Yes

PLOS authors have the option to publish the peer review history of their article (what does this mean? ). If published, this will include your full peer review and any attached files.

**Do you want your identity to be public for this peer review?** For information about this choice, including consent withdrawal, please see our Privacy Policy .

Reviewer #2: No

Reviewer #3: No

**Data Deposition**

http://datadryad.org/submit?journalID=pgenetics&manu=PGENETICS-D-25-00446R1

**Press Queries**

---

## [Editor Report · Acceptance letter]

PGENETICS-D-25-00446R1

Tissue-specific consequences of tag fusions on protein expression in transgenic mice

Dear Dr Wood,

We are pleased to inform you that your manuscript entitled "Tissue-specific consequences of tag fusions on protein expression in transgenic mice" has been formally accepted for publication in PLOS Genetics! Your manuscript is now with our production department and you will be notified of the publication date in due course.

With kind regards,

Benedek Toth

PLOS Genetics

On behalf of:
